# Membership Inference Attacks on Deep Regression Models for Neuroimaging

**Umang Gupta**[1]                   UMANGGUP@USC.EDU

**Dmitris Stripelis**[1]                  STRIPELI@ISI.EDU

**Pradeep K. Lam**[2]                  PRADEEPL@USC.EDU

**Paul M. Thompson**[2]                 PTHOMP@USC.EDU

**José Luis Ambite**[1]                  AMBITE@ISI.EDU

**Greg Ver Steeg**[1]                   GREGV@ISI.EDU

[1]*Information Sciences Institute, University of Southern California*

[2]*Imaging Genetics Center, Mark and Mary Stevens Institute for Neuroimaging and Informatics, Keck School of Medicine, University of Southern California*

## Abstract

Ensuring the privacy of research participants is vital, even more so in healthcare environments. Deep learning approaches to neuroimaging require large datasets, and this often necessitates sharing data between multiple sites, which is antithetical to the privacy objectives. Federated learning is a commonly proposed solution to this problem. It circumvents the need for data sharing by sharing parameters during the training process. However, we demonstrate that allowing access to parameters may leak private information even if data is never directly shared. In particular, we show that it is possible to infer if a sample was used to train the model given only access to the model prediction (black-box) or access to the model itself (white-box) and some leaked samples from the training data distribution. Such attacks are commonly referred to as *Membership Inference attacks*. We show realistic Membership Inference attacks on deep learning models trained for 3D neuroimaging tasks in a centralized as well as decentralized setup. We demonstrate feasible attacks on brain age prediction models (deep learning models that predict a person's age from their brain MRI scan). We correctly identified whether an MRI scan was used in model training with a 60% to over 80% success rate depending on model complexity and security assumptions.

## 1. Introduction

Machine learning's endless appetite for data is increasingly in tension with the desire for data privacy. Privacy is a highly significant concern in medical research fields such as neuroimaging, where information leakage may have legal implications or severe consequences on individuals' quality of life. The *Health Insurance Portability and Accountability Act 1996* (HIPAA) (Centers for Medicare & Medicaid Services, 1996) protects the health records of an individual subject, as well as data collected for medical research. Privacy laws have spurred research into anonymization algorithms. One such example is algorithms that remove facial information from MRI scans (Bischoff-Grethe et al., 2007; Schimke and Hale, 2011; Milchenko and Marcus, 2013).

While there are laws and guidelines to control private data sharing, model sharing or using models learned from private data may also leak information. The risk to participants' privacy, even when only summary statistics are released, has been demonstrated and widely discussed in the field of genome-wide association studies (Homer et al., 2008; Craig et al., 2011). In a similar spirit, a neural network model learned from private data can be seen as a summary statistic of the data, and private information may be extracted from it. To demonstrate the feasibility of information leakage, we study the problem of extracting information about individuals from a model trained on the 'brain age prediction' regression task using neuroimaging data. Brain age is the estimate of a person's age from their brain MRI scan, and it is a commonly used task for benchmarking machine learning algorithms.

In particular, we study attacks to infer which samples or records were used to train the model. These are called *Membership Inference attacks* (Shokri et al., 2017; Nasr et al., 2019). An adversary may infer if an individual's data was used to train the model, thus violating privacy through these attacks. Consider a hypothetical example, where some researchers released a neural network trained with scans of participants in a depression study. An adversary with access to the individual's scan and the model may identify if they participated in the study, revealing information about their mental health, which can have undesirable consequences.

Previous work on membership inference attacks focus on supervised *classification* problems, often exploiting the model's over-confidence on the training set and the high dimensionality of the probability vector (Shokri et al., 2017; Salem et al., 2019; Pyrgelis et al., 2017). Our work demonstrates membership inference attacks on *regression* models trained to predict a person's age from their brain MRI scan (brain age) under both white-box and black-box setups. We simulate attacks on the models trained under centralized as well as distributed, federated setups. We also demonstrate a strong empirical connection between overfitting and vulnerability of the model to membership inference attacks.

## 2. Related Work & Background

### 2.1. BrainAGE Problem

*Brain age* is an estimate of a person's age from a structural MRI scan of their brain. The difference between a person's true chronological age and the predicted age is a useful biomarker for early detection of various neurological diseases (Franke and Gaser, 2019) and the problem of estimating this difference is defined as the Brain Age Gap Estimation (BrainAGE) problem. Brain age prediction models are trained on brain MRIs of healthy subjects to predict the chronological age. A higher gap between predicted and chronological age is often considered an indicator of accelerated aging in the subject, which may be a prodrome for neurological diseases. To predict age from raw 3D-MRI scans, many recent papers have proposed using deep learning (Feng et al., 2020; Gupta et al., 2021; Stripelis et al., 2021; Peng et al., 2021; Bashyam et al., 2020; Lam et al., 2020). To simulate attacks on models trained centrally and distributively, we employ trained networks and training setups recently proposed in Gupta et al. (2021) and Stripelis et al. (2021), respectively. Although there is some controversy over the interpretation of BrainAGE (Butler et al., 2020; Vidal-Pineiro et al., 2021), we emphasize that we are only using BrainAGE as a representative problem in neuroimaging that benefits from deep learning.

## 2.2. Federated Learning

In traditional machine learning pipelines, data originating from multiple data sources must be aggregated at a central repository for further processing and analysis. Such an aggregation step may incur privacy vulnerabilities or violate regulatory constraints and data sharing laws, making data sharing across organizations prohibitive. To address this limitation, *Federated Learning* was recently proposed as a distributed machine learning paradigm that allows institutions to collaboratively train machine learning models by relaxing the need to share private data and instead push the model training locally at each data source (McMahan et al., 2017; Yang et al., 2019; Kairouz and McMahan, 2021). Even though Federated Learning was originally developed for mobile and edge devices, it is increasingly applied in biomedical and healthcare domains due to its inherent privacy advantages (Lee et al., 2018; Sheller et al., 2018; Silva et al., 2019; Rieke et al., 2020; Silva et al., 2020).

Depending on the communication characteristics between the participating sources, different federated learning topologies can be discerned (Yang et al., 2019; Bonawitz et al., 2019; Rieke et al., 2020; Bellavista et al., 2021) — *star* and *peer-to-peer* being the most prominent. In a star topology (Sheller et al., 2018; Li et al., 2019, 2020; Stripelis et al., 2021), the execution and training coordination across sources is realized by a trusted centralized entity, the *federation controller*, which is responsible for shipping the global or *community model* to participating sites and aggregating the local models. In peer-to-peer (Roy et al., 2019) topologies, the participating sites communicate directly with each other without requiring a centralized controller. We focus on the star federated learning topology.

In principle, at the beginning of the federation training, every participating data source or *learner* receives the community model from the federation controller, trains the model independently on its local data for an assigned number of iterations, and sends the locally trained parameters to the controller. The controller computes the new community model by aggregating the learners' parameters and sends it back to the learners to continue training. We refer to this synchronization point as a *federation round*. After repeating multiple federation rounds, the jointly learned community model is produced as the final output.

## 2.3. Membership Inference Attacks

Membership inference attacks are one of the most popular attacks to evaluate privacy leakage in practice (Jayaraman and Evans, 2019). The malicious use of trained models to infer which subjects participated in the training set by having access to some or all attributes of the subject is termed as *membership inference attack* (Shokri et al., 2017; Nasr et al., 2019). These attacks aim to infer if a record (a person's MRI scan in our case) was used to train the model, revealing information about the subject's participation in the study, which could have legal implications. These attacks are often distinguished by the access to the information that the adversary has (Nasr et al., 2019). Most successful membership inference attacks in the deep neural network literature require access to some parts of the training data or at least some samples from the training data distribution (Salem et al., 2019; Pyrgelis et al., 2017; Truex et al., 2018). *White-box attacks* assume that the attacker is also aware of the training procedure and has access to the trained model parameters, whereas *Black-box attacks* only assume unlimited access to an API that provides the output of the model (Leino and Fredrikson, 2020; Nasr et al., 2019).

Creating efficient membership inference attacks with minimal assumptions and information is an active area of research (Choo et al., 2020; Jayaraman et al., 2020; Song and Mittal, 2020). However, our work is focused on demonstrating the vulnerability of deep neural networks to membership inference attacks in the federated as well as non-federated setup. Therefore, we make straightforward assumptions and assume somewhat lenient access to information. Our attack models are inspired by Nasr et al. (2019); Shokri et al. (2017), and we use similar features such as gradients of parameters, activations, predictions, and labels to simulate membership inference attacks. In particular, we learn deep binary classifiers to distinguish training samples from unseen samples using these features.

In the case of federated learning, each learner receives model parameters and has some private training data. Thus, any learner is capable of launching white-box attacks. Moreover, in this scenario, the learner has access to the community models received at each federation round. When simulating membership attacks on federated models, we simulate attacks from the learners' perspective by training on learners' private data and the task is to identify other learners' subjects. In the case of models trained via centralized training, we assume that the adversary can access some public training and test samples. We simulate both white-box and black-box attacks in this case.

## 3. Setup

### 3.1. Trained Models for Predicting Brain Age

We use models trained to predict brain age from structural MRIs to demonstrate vulnerability to membership inference attacks. We show successful attacks on `3D-CNN` and `2D-slice-mean` models. The neural network architectures are summarized in Appendix A.3. For centralized training, we use the same dataset and training setup as Gupta et al. (2021) and for federated training, we use the same training setup and dataset as Stripelis et al. (2021) (see Appendices A.1 and A.2). In the latter, the authors simulate different federated training environments by considering diverse amounts of records (i.e., Uniform and Skewed) and varying subject age distribution across learners (i.e., IID and non-IID). All models are trained on T1 structural MRI scans of healthy subjects from the UK Biobank dataset (Miller et al., 2016) with the same pre-processing as Lam et al. (2020). See Appendix A for more details regarding the dataset, data distribution, and training setup.

### 3.2. Attack Setup

As discussed in Section 2.3, attackers may have access to some part of the training set and additional MRI samples that were not used for training, referred hereafter as the *unseen set*. We train a binary classifier to distinguish if the sample was part of the training set (see Appendix C for classifier architecture details). We study effectiveness of different features for the attacks in Section 4.1.

In the case of models trained via centralized training, the attack models are trained on a balanced training set using 1500 samples from both training and unseen sample set[1]. For testing, we create a balanced set from the remaining train and unseen set — 694

---

1. In the implementation, the unseen set is the same as the test dataset used to evaluate the brain age model. The unseen set and the training set are IID samples from the same distribution.

samples each and report accuracy as the vulnerability measure. To attack models trained via federated learning, we consider each learner as the attacker. Thus, the attacker is trained on its private dataset and some samples from the unseen set that it may have. This way, we created a balanced training set of up to 1000[2] samples from training and unseen set each. Unlike centralized setup, the distribution of the unseen set and training set that the attacker model is trained on could be different, particularly in non-IID environments. In this scenario, the attacks are made on the private data of other learners. Thus, we report the classifier's accuracy on the test set created from the training sample of the learner being attacked and new examples from the unseen set.

## 4. Results

We simulate membership inference attacks on both centralized and federation trained models for the BrainAGE problem. We report results on models trained centrally in Section 4.1 and distributively in Section 4.2. Conventional deep learning models are trained using gradient descent. Thus, the gradient of parameters w.r.t. loss computed from a trained model are likely to be lower for the training set than the unseen set. We evaluate features derived from gradients, activation, errors, and predictions of the trained model to train the binary classifier and study their effectiveness in Section 4.1. The main task is to identify if a sample belonged to the training set. We report the accuracy of correct identification on a test set created from the training and the unseen sample sets that were not used to train the attack model but used for training and evaluating the brain age models.

### 4.1. Membership Inference Attacks on Centralized Training

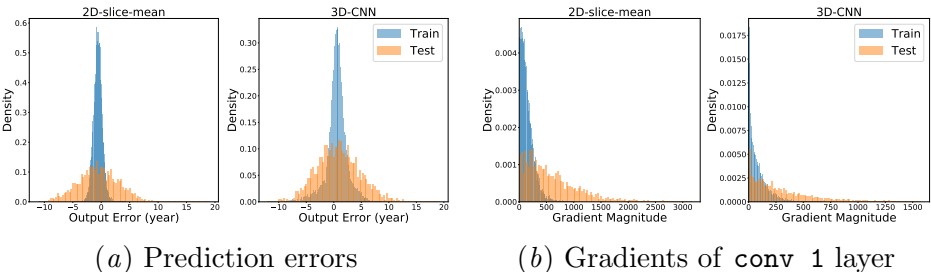

($a$) Prediction errors        ($b$) Gradients of `conv 1` layer

Figure 1: Distribution of prediction error and gradient magnitudes from the trained models.

Table 1 summarizes the results of simulating membership attacks with various features. As apparent from Figure 1($a$), test and train samples have different error distributions due to the inherent tendency of deep neural networks to overfit on the training set (Zhang et al., 2017). Consequently, the error is a useful feature for membership inference attacks. Error is the difference between prediction and label, and using prediction and label as two separate features produced even stronger attacks, as indicated by higher membership attack accuracies. One of the reasons for this could be that the model overfits more for some age

---

2. In the case of Skewed & non-IID environment, some learners had less than 1000 training samples. As a result, the attack model had to be trained with fewer samples.

| Features | 3D-CNN | 2D-slice-mean |
|---|---|---|
| activation | 56.63 | - |
| error | $59.90 \pm 0.01$ | $74.06 \pm 0.00$ |
| gradient magnitude | $72.60 \pm 0.45$ | $78.34 \pm 0.17$ |
| gradient (conv 1 layer) | $71.01 \pm 0.64$ | $80.52 \pm 0.40$ |
| gradient (output layer) | $76.65 \pm 0.44$ | $82.16 \pm 0.29$ |
| gradient (conv 6 layer) | $76.96 \pm 0.57$ | $82.89 \pm 0.83$ |
| prediction + label | $76.45 \pm 0.20$ | $81.70 \pm 0.29$ |
| prediction + label + gradient (conv 6 + output) | $78.05 \pm 0.47$ | $83.04 \pm 0.50$ |

Table 1: Membership inference attack accuracies on centrally trained models (averaged over 5 attacks). Details about conv 1, output and conv 6 layers are provided in Appendix A.3.

groups. Using true age information (label) would enable the attack model to find these age groups, resulting in higher attack accuracy.

Attacks made using error or prediction, and label are black-box attacks. A white-box attacker may also utilize more information about the models' internal workings like the gradients, knowledge about loss function, training algorithm, etc. Deep learning models are commonly trained until convergence using some variant of gradient descent. The convergence is achieved when the gradient of loss w.r.t parameters on the training set is close to 0. As a result, gradient magnitudes are higher or similar for unseen samples than training samples (see Figure 1(b)). Therefore, we used the gradient magnitude of each layer as a feature, resulting in attack accuracy of 72.6 and 78.34 for 3D-CNN and 2D-slice-mean models, respectively. Finally, we simulated attacks using gradients of parameters at different layers[3]. We find that parameter-gradients of layers closer to the output layer (i.e., conv 6, output layers) are more effective compared to the gradients of layers closer to the input (conv 1). Preliminary results hinted that activations do not provide much information to attack the models. So, we did not simulate attacks on the 2D-slice-mean models with activations as features. The best attack accuracies of 78.05 and 83.04 for attacking 3D-CNN and 2D-slice-mean model were achieved by using prediction, labels, and gradients of parameters close to the output layer. Successful membership inference attacks demonstrated in this section accessed samples from the training set, which is limiting. In Appendix E, we discuss attacks accessing only the training set distribution and not the training samples.

## 4.2. Membership Inference Attacks on Federated Training

We consider three different federated learning environments consisting of 8 learners and investigate cases where malicious learners attack the community model. The community model is the aggregated result of learners' local models and a malicious learner may use it to extract information about other learners' training samples. In this scenario, a malicious learner can learn an attack model by leveraging its access to the community models of all federation rounds and its local training dataset; we simulate attacks using this information

---

3. We consider layers close to the input or output layers as these have fewer parameters, and attack models are easily trained. Intermediate layers had more parameters, making it hard to learn the attack model.

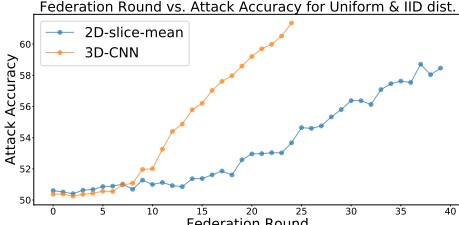

Figure 2: Increasing attack vulnerability per federation round.

| Data distribution | 3D-CNN | 2D-slice-mean |
|---|---|---|
| Uniform & IID | 60.06 (56) | 58.11 (56) |
| Uniform & non-IID | 61.00 (28) | 60.28 (29) |
| Skewed & non-IID | 64.12 (25) | 63.81 (24) |

Table 2: Average attack accuracies on federation trained models. Numbers in parentheses indicate median successful attacks over 5 multiple runs.

(see also Section 3.2). The model vulnerability is likely to increase with more training iterations and hence we used features derived from the community models received during the last five federation rounds, and each learner uses its private samples to learn the attack model. Each learner may try to do membership inference attacks on any of the other seven learners, resulting in 56 possible attack combinations. An attack is considered successful if accuracy is more than 50%, which is the random prediction baseline.

Table 2 shows the average accuracy of successful attacks and the total number of successful attack instances of learner-attacker pairs (in parentheses) across all possible learner-attacker pairs (56 in total). For a more detailed analysis on a per-learner basis, see Appendix B. We empirically observed that the success rate of the attacks is susceptible to data distribution shifts. In particular, distribution shift agnostic features like gradient magnitudes can lead to more successful attacks (count wise) when data distribution across learners differs. For the results shown in Table 2 and Figure 2, we used all available features (i.e., gradient magnitudes, predictions, labels, and gradients of last layers).

We also observe that the overall attack accuracies are lower than the centralized counterpart discussed in Section 4.1. This drop can be attributed to the following: a) As we show in Section 4.3, attack accuracies are highly correlated with overfitting. Federated learning provides more regularization than centralized training and reduces overfitting but does not eliminate the possibility of an attack. b) Federated models are slow to train, but as the model is trained for more federation rounds, the vulnerability increases (see Figure 2). Moreover, Table 2 only presents an average case view of the attacks and we observe that the attack performance depends on the data distribution of the learner-attacker pair. When the local data distribution across learners is highly diverse, i.e., *Skewed & non-IID* attack accuracies can be as high as 80% for specific learner-attacker pairs (see Appendix B).

### 4.3. Possible Defenses

Various approaches have been proposed to mitigate the membership inference attacks directly. These approaches are based on controlling overfitting (Truex et al., 2018; Salem et al., 2019) and training data memorization (Jha et al., 2020) or adversarial training (Nasr et al., 2018). We evaluate differentially private machine learning as one of the defenses.

Differential privacy (Dwork and Roth, 2014) is often touted as a panacea for all privacy-related problems. We evaluate the effect of training models with privacy guarantees on membership inference attacks and model performance, measured as mean absolute error

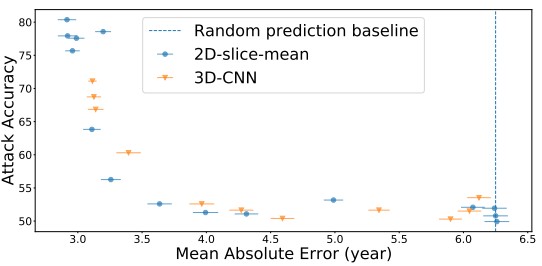
(a) Attack accuracy vs. Model performance

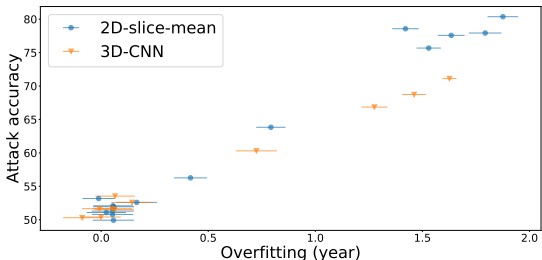
(b) Attack accuracy vs. Overfitting

Figure 3: Differential privacy reduces membership inference attacks. Figure (b) shows that the effectiveness of membership inference attack is correlated with overfitting. Error bars are generated by bootstrapping the test set 5 times using 1000 samples. Results with $R^2$ as the measure of model performance are shown in Appendix D.

in the centralized setup. To train the models with differential privacy, we used `DP-SGD` algorithm of Abadi et al. (2016) which works by adding Gaussian noise to the gradient updates from each sample[4]. We varied the noise magnitude to achieve different points on the trade-off curves of Figure 3. Models trained with differential privacy significantly reduce attack accuracy, but this is achieved at the cost of a significant drop in model performance (Figure 3(a)). We visualize the relation between overfitting, measured by train and test performance difference, and attack vulnerability of the models trained with differential privacy in Figure 3(b). We see that overfitting is highly correlated with attack accuracy, indicating that these attacks may be prevented by avoiding overfitting up to some extent.

## 5. Discussion

While deep learning presents great promise for solving neuroimaging problems, it also brings new challenges. Deep learning is intrinsically data-hungry, but the bulk of neuroimaging data is distributed around the world in private repositories. With classic machine learning approaches like linear regression, model sharing and meta-analysis could be used to pool insights without sharing data. Unfortunately, neural networks are capable of completely memorizing training data, so that sharing a model may be just as bad as sharing the private data itself. In this paper, we demonstrated a practical proof-of-concept attack for extracting private information from neural networks trained on neuroimaging data. We showed that attacks with a high success rate persist under various settings, including a realistic, distributed, federated learning scheme explicitly designed to protect private information. Although concerning, our preliminary study of attacks and defenses suggest benefits to solving this problem that go beyond data privacy. Because attacks exploit differences in model performance on training data and unseen test data, a successful defense must also lead to more robust neuroimaging models whose out-of-sample performance does not significantly differ from in-sample performance. Hence, even if data privacy were not a concern, further study of protection against membership attacks may inspire neuroimaging models that generalize better to new patients.

---

4. For a brief description of differential privacy and details of differential private training, see Appendix D.

## Acknowledgments

This research was supported by DARPA contract HR0011-2090104. PL and PT were supported by the NIH under grant U01 AG068057 and by a research grant from Biogen, Inc. This research has been conducted using the UK Biobank Resource under Application Number 11559.

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

## Appendix A. Brain Age Model, Training and Dataset Details

In both federated and centralized setups, we used T1 structural MRI scans of healthy subjects from the UK Biobank dataset (Miller et al., 2016) for brain age prediction. All the scans were preprocessed with the same technique as Lam et al. (2020), resulting in final images with dimensions $91 \times 109 \times 91$. Here we briefly describe the relevant details. We refer the reader to Gupta et al. (2021) and Stripelis et al. (2021) for full details.

### A.1. Centralized Training Setup

| Model | Train | Test | Validation |
|---|---|---|---|
| 3D-CNN | 1.39 | 3.13 | 3.09 |
| 2D-slice-mean | 0.77 | 2.88 | 2.92 |

Table 3: Mean absolute errors (year) for train, test and validation set in the centralized setup.

To simulate attacks on centrally trained deep neural network models, we adopted the pretrained models from Gupta et al. (2021). The authors selected a subset of healthy 10,446 subjects from 16,356 subjects in the UK Biobank dataset to create a training, validation, and test set of size 7,312, 2,194, and 940, respectively, with a mean chronological age of 62.6 and standard deviation of 7.4 years. Gupta et al. (2021) proposed novel 2D-slice-based architectures to improve brain age prediction. Their architectures used 2D convolutions to encode the slices along the sagittal axis and aggregated the resultant embeddings through permutation invariant operations. In our work, we use the 2D-slice-mean model, which demonstrated the best performance in their study, and a conventional 3D-CNN model, which is often used to process MRI scans (Peng et al., 2021; Cole et al., 2017). The architecture diagram of both the models are shown in Figure 6 and discussed in Section A.3.

For the brain age problem, the performance is measured as the mean absolute error (MAE) between the predicted and true age on the held-out test set. In Gupta et al. (2021), the models were trained for 100 epochs, and the best model was selected based on the performance on the validation set. The membership inference attacks that we investigate in this work are evaluated over the models produced at the end of the $100^{th}$ epoch. Table 3 shows performance of these models, i.e., MAE on train, test and validation sets at the end of $100^{th}$ epoch.

## A.2. Federated Training Setup

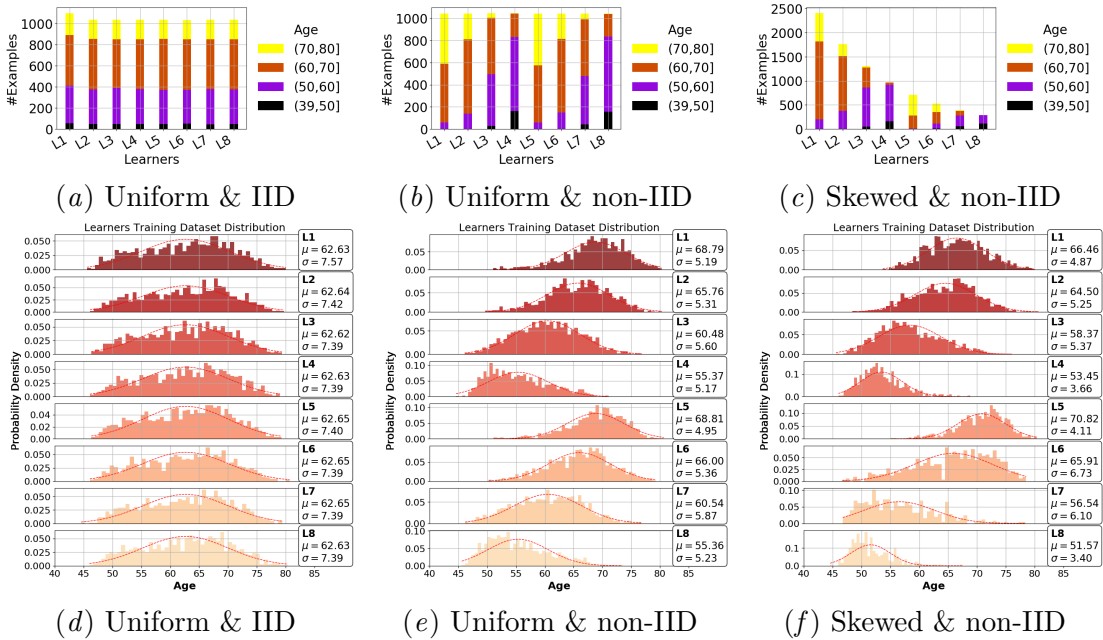

Figure 4: The UK Biobank data distribution across 8 learners for the three federated learning environments. Figures $(a)$, $(b)$ and $(c)$ present the amount of data per age range bucket (i.e., $[39 - 50), [50 - 60), [60 - 70), [70 - 80)$) per learner. Figures $(d)$, $(e)$ and $(f)$ present the age range distribution (mean $\mu$ and standard deviation $\sigma$) per learner. Figures are reproduced from Stripelis et al. (2021).

| Model | Uniform & IID | | Uniform & non-IID | | Skewed & non-IID | |
|---|---|---|---|---|---|---|
| | Train | Test | Train | Test | Train | Test |
| 3D-CNN | 2.16 | 3.01 | 3.41 | 3.81 | 2.83 | 3.47 |
| 2D-slice-mean | 1.81 | 2.76 | 2.40 | 2.98 | 2.42 | 3.10 |

Table 4: Mean absolute errors (year) for training, and testing set for different environments in the federated setup.

To simulate membership inference attacks on models trained in federated learning environment, we used the pretrained models, dataset, and training setup of Stripelis et al. (2021). In particular, the investigated learning environments consist of 8 learners with homogeneous computational capabilities (8 GeForce GTX 1080 Ti graphics cards with 10 GB RAM each) and heterogeneous local data distributions. With respect to the UK Biobank dataset, the 10,446 subject records were split into 8,356 train and 2,090 test samples. In particular, three representative federated learning environments were generated with diverse amounts of records (i.e., Uniform and Skewed) and subject age range distribution across learners (i.e., IID and non-IID). All these environments are presented in Figure 4.

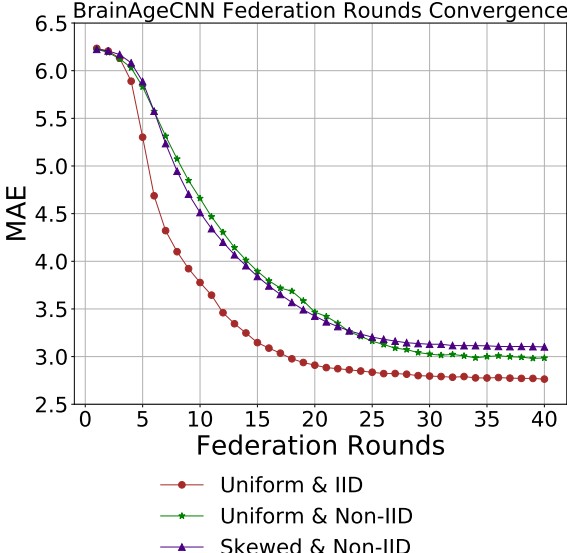

Figure 5: Learning curve (test performance) for `2D-slice-mean` model across different federated learning environments. The model is evaluated at each federation round for the brain age prediction problem. The more non-IID and unbalanced the data distribution is, the harder it is for the federation model to converge.

To perform our attacks, we considered the community models received by each learner in all federation rounds. Specifically, we used the pretrained `3D-CNN` community models from Stripelis et al. (2021), which were trained for 25 federation rounds, and every learner performed local updates on the received community model parameters for 4 epochs in each round. To train the `2D-slice-mean` federation model, we emulate a similar training setup for 40 federation rounds. For both federated models, the solver of the local objective is SGD, the batch size is equal to 1, the learning rate is equal to $5e^{-5}$ and every learner used all its local data during training, without reserving any samples for validation. Finally, at every federation round all local models are aggregated using the Federated Average (FedAvg) aggregation scheme (McMahan et al., 2017). The convergence of the `2D-slice-mean` federated model for the three federated learning environments is shown in Figure 5 and the performance of the final community models for each learning environment is summarized in Table 4.

### A.3. `3D-CNN` and `2D-slice-mean` model architecture

`3D-CNN`: Figure 6(a) describes the architecture for the `3D-CNN` model. `3D-CNN` uses 5 convolutional blocks consisting of 3D-convolution layers with 32, 64, 128, 256 and 256 filters. Each convolutional layer is followed by 3D max-pooling, 3D instance norm and `ReLU` non-linearity operations. The resulting activations from these are passed through a 64 filter convolutional layer of kernel size 1, average pooled and passed through another 3D-convolutional layer of kernel size 1 to produce the 1 dimensional brain age output.

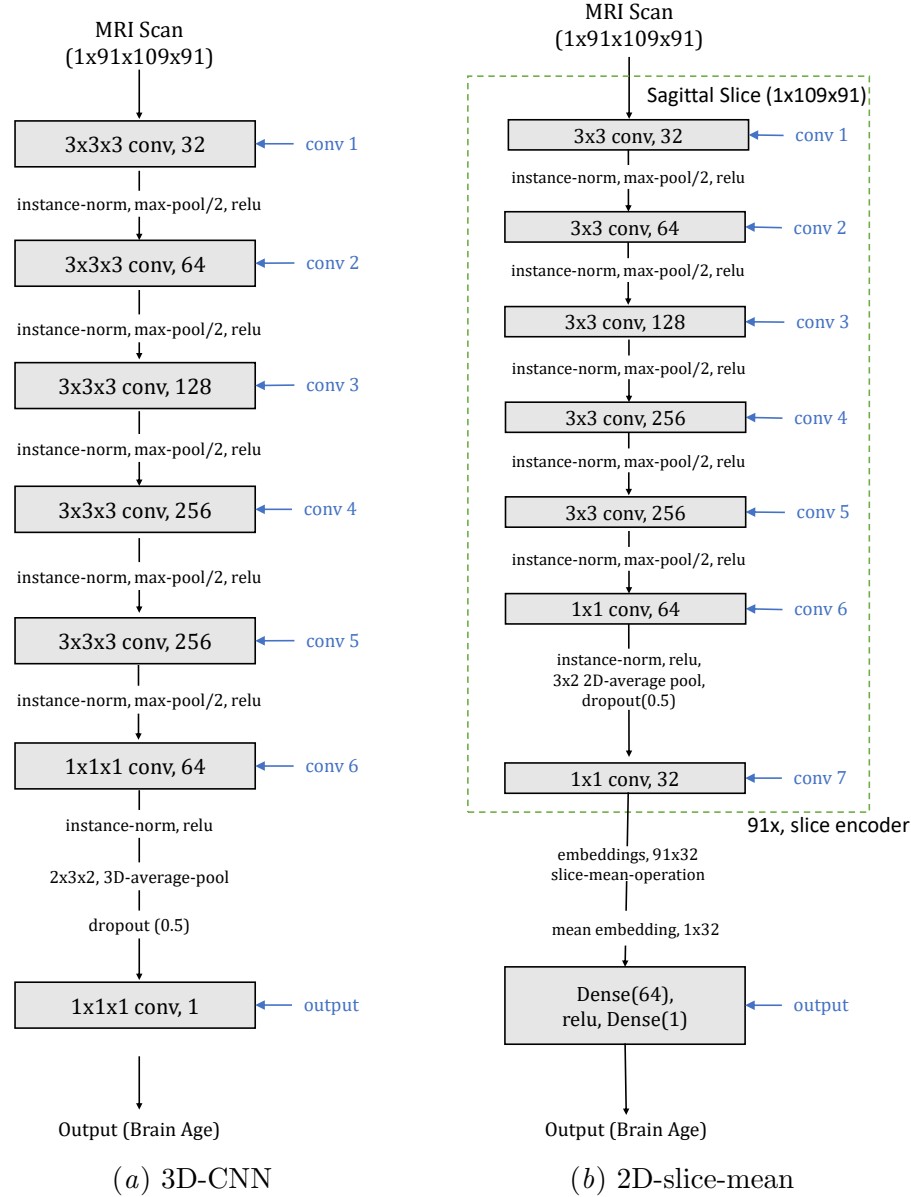

(a) 3D-CNN          (b) 2D-slice-mean

Figure 6: Neural network architectures for brain age prediction. Gray blocks indicate trainable modules and non-parametric operations are indicated on the arrows. Groups of parameters are labelled for the ease of reference.

**2D-slice-mean:** Figure 6(*b*) describes the architecture of the 2d-slice-mean models. This architecture encodes each slice along the sagittal dimensional using a slice encoder. The slice encoder is similar to the 3D-CNN model but uses the 2D version of all the operations. Ultimately, all the slices are projected to a 32-dimensional embedding. The slice-mean operation aggregates these 32-dimensional embeddings via mean operation, which are then passed through feed-forward layers to output the brain age.

## Appendix B. Detailed Results of Membership Inference Attacks on Federated Learning

| Features | 3D-CNN | | | 2D-slice-mean | | |
|---|---|---|---|---|---|---|
| | D1 | D2 | D3 | D1 | D2 | D3 |
| Set 1 | $60.08 \pm 0.06$ (56) | $57.75 \pm 0.15$ (30) | $59.01 \pm 0.34$ (29) | $58.15 \pm 0.07$ (56) | $55.27 \pm 0.03$ (37) | $58.55 \pm 0.38$ (24) |
| Set 2 | $60.09 \pm 0.06$ (56) | $59.97 \pm 0.29$ (30) | $63.59 \pm 0.35$ (26) | $58.04 \pm 0.23$ (56) | $60.41 \pm 0.22$ (29) | $63.73 \pm 0.27$ (25) |
| Set 3 | $60.06 \pm 0.04$ (56) | $61.00 \pm 0.47$ (28) | $64.12 \pm 0.52$ (25) | $58.11 \pm 0.22$ (56) | $60.28 \pm 0.73$ (29) | $63.81 \pm 0.55$ (24) |

Table 5: Average membership inference attack accuracies on models trained using federated learning across all environments using different feature sets. Standard deviations are reported over 5 runs. Number in parentheses indicate the median total number of successful attacks over 5 runs.

**Table Legend:**
    D1: Uniform & IID data distribution
    D2: Uniform & non-IID data distribution
    D3: Skewed & non-IID data distribution
    Set 1: Gradient magnitude
    Set 2: Gradient magnitude + prediction + label
    Set 3: Gradient magnitude + prediction + label + gradient (conv 6 + output)

In Section 4.2, we discussed summary results of attacks on models trained via federated learning. Here, we provide a more detailed analysis of the attack results. Table 5 compares the attack performance of different feature sets. We observe that in federated environments with similar data sizes and homogeneous data distribution, i.e., Uniform & IID, all attacks succeeded. However, when the local data size and the data distribution across learners are heterogeneous, the total number of successful attacks decreases, indicating that attacks are sensitive to data distribution. It is interesting to note that even though using only magnitudes as a feature resulted in poor average attack performance, these features may be more robust to distribution shift and have more successful attacks in some cases. Investigating and designing more robust features for membership inference attacks may lead to even more adverse attacks.

Tables 6 and 7 visualize the attack results on a per learner basis. Each row indicates the attacker, and the column indicates the results of the attack on the attacked learner. We observe that the attack performance is correlated with the distribution similarity. For

**3D-CNN**

**Environment: Uniform & IID**

| | L1 | L2 | L3 | L4 | L5 | L6 | L7 | L8 |
|---|---|---|---|---|---|---|---|---|
| L1 | | 60.62±0.66 | 60.71±0.61 | 59.85±0.69 | 59.89±0.72 | 59.17±0.58 | 60.64±0.46 | 60.15±0.51 |
| L2 | 59.68±0.20 | | 60.57±0.69 | 60.02±0.67 | 59.86±0.58 | 59.10±0.32 | 60.63±0.53 | 60.15±0.45 |
| L3 | 59.78±0.15 | 60.59±0.41 | | 60.18±0.46 | 60.19±0.48 | 59.55±0.13 | 60.63±0.28 | 60.48±0.28 |
| L4 | 59.92±0.26 | 60.29±0.40 | 60.89±0.25 | | 60.14±0.32 | 59.45±0.29 | 60.73±0.24 | 60.58±0.20 |
| L5 | 59.86±0.33 | 60.22±0.25 | 60.85±0.13 | 59.94±0.36 | | 59.41±0.26 | 60.48±0.33 | 60.55±0.34 |
| L6 | 59.98±0.30 | 59.98±0.35 | 60.56±0.41 | 59.56±0.47 | 60.07±0.57 | | 60.70±0.47 | 60.32±0.48 |
| L7 | 60.08±0.45 | 60.05±0.38 | 60.81±0.37 | 59.77±0.18 | 60.00±0.30 | 59.28±0.31 | | 60.29±0.35 |
| L8 | 59.09±0.30 | 59.13±0.24 | 60.15±0.32 | 59.28±0.39 | 59.34±0.12 | 58.98±0.24 | 59.66±0.25 | |

**Environment: Uniform & non-IID**

| | L1 | L2 | L3 | L4 | L5 | L6 | L7 | L8 |
|---|---|---|---|---|---|---|---|---|
| L1 | | 56.20±1.44 | 38.34± 0.51 | 33.06± 0.73 | 72.72±0.36 | 55.92±1.22 | 38.25± 0.33 | 32.99± 0.71 |
| L2 | 56.36±2.20 | | 52.80±1.09 | 39.70± 1.32 | 56.61±2.41 | 63.28±0.12 | 51.03±1.11 | 40.20± 1.42 |
| L3 | 32.76± 0.95 | 49.15± 0.83 | | 66.09±1.50 | 32.42± 0.77 | 49.63± 0.77 | 63.43±0.49 | 66.61±1.19 |
| L4 | 32.05± 0.29 | 41.50± 0.10 | 59.98±0.14 | | 32.45± 0.25 | 43.12± 0.19 | 61.45±0.11 | 69.86±0.21 |
| L5 | 72.28±0.19 | 54.02±0.15 | 37.46± 0.34 | 33.53± 0.23 | | 54.09±0.21 | 37.43± 0.10 | 33.59± 0.17 |
| L6 | 59.79±1.22 | 62.71±0.56 | 51.31±0.95 | 37.86± 0.84 | 59.59±1.45 | | 49.37± 0.69 | 38.51± 0.86 |
| L7 | 31.96± 0.76 | 49.26± 0.57 | 64.98±0.20 | 66.40±1.06 | 31.46± 0.80 | 49.72± 0.54 | | 67.11±0.65 |
| L8 | 32.88± 0.26 | 42.03± 0.79 | 59.64±1.24 | 69.93±0.24 | 33.26± 0.10 | 43.15± 0.85 | 61.23±0.65 | |

**Environment: Skewed & non-IID**

| | L1 | L2 | L3 | L4 | L5 | L6 | L7 | L8 |
|---|---|---|---|---|---|---|---|---|
| L1 | | 66.85±0.38 | 40.96± 0.26 | 30.86± 0.65 | 64.44±1.39 | 59.41±0.33 | 40.13± 0.25 | 31.38± 1.10 |
| L2 | 67.90±0.60 | | 50.38±0.50 | 35.40± 1.33 | 48.98± 0.86 | 54.98±0.28 | 43.17± 0.52 | 32.34± 1.67 |
| L3 | 40.26± 0.50 | 50.31±0.63 | | 62.31±1.09 | 33.61± 0.79 | 46.84± 0.45 | 64.99±0.60 | 59.93±1.92 |
| L4 | 36.86± 0.08 | 40.38± 0.20 | 60.66±0.76 | | 35.99± 0.17 | 41.34± 0.22 | 65.98±0.36 | 83.10±0.17 |
| L5 | 58.36±0.65 | 51.96±1.05 | 36.82± 1.82 | 34.29± 1.80 | | 58.98±0.16 | 36.45± 1.05 | 32.45± 1.59 |
| L6 | 67.88±2.40 | 60.44±4.63 | 35.74± 2.27 | 28.04± 2.37 | 70.74±1.33 | | 34.40± 1.07 | 26.76± 2.75 |
| L7 | 35.89± 0.54 | 42.13± 0.64 | 64.86±0.43 | 78.70±0.32 | 33.38± 0.27 | 43.37± 0.35 | | 81.14±0.26 |
| L8 | 39.64± 0.19 | 40.89± 0.12 | 53.93±0.43 | 76.36±0.18 | 39.48± 0.23 | 41.97± 0.10 | 62.53±0.16 | |

Table 6: Matrix of the membership inference attack accuracy on a per learner basis for the 3D-CNN model across every federated learning environment. Rows are the attacking learner and columns are the attacked learner. Colored cells indicate successful attacks and more heated cells specify higher attack accuracies. The results are over 5 runs.

**2D-slice-mean**

Environment: Uniform & IID

| | L1 | L2 | L3 | L4 | L5 | L6 | L7 | L8 |
|---|---|---|---|---|---|---|---|---|
| L1 | | 58.63±1.80 | 57.18±2.27 | 57.15±1.61 | 57.41±1.07 | 56.89±1.43 | 57.02±1.19 | 57.77±1.54 |
| L2 | 57.49±0.70 | | 58.74±1.00 | 57.93±0.89 | 57.82±0.79 | 58.15±0.48 | 57.63±0.95 | 59.32±1.15 |
| L3 | 58.22±0.68 | 60.61±0.66 | | 58.81±0.38 | 58.95±0.58 | 58.70±0.45 | 58.66±0.61 | 60.29±0.44 |
| L4 | 57.07±0.81 | 58.69±1.14 | 57.18±0.94 | | 57.43±0.70 | 58.20±0.90 | 56.34±0.99 | 58.01±0.95 |
| L5 | 57.69±1.12 | 60.28±1.65 | 58.64±1.98 | 58.26±1.09 | | 57.99±1.57 | 57.85±1.64 | 59.26±1.56 |
| L6 | 56.35±0.63 | 58.94±0.80 | 56.85±1.06 | 57.20±0.64 | 56.47±0.58 | | 55.98±1.04 | 58.34±1.06 |
| L7 | 58.00±0.44 | 60.62±0.48 | 59.24±0.90 | 58.65±0.30 | 59.14±0.65 | 58.31±1.15 | | 59.91±0.52 |
| L8 | 57.20±1.25 | 59.85±1.05 | 57.53±1.28 | 57.64±1.18 | 57.58±0.98 | 56.72±1.09 | 57.24±1.12 | |

Environment: Uniform & non-IID

| | L1 | L2 | L3 | L4 | L5 | L6 | L7 | L8 |
|---|---|---|---|---|---|---|---|---|
| L1 | | 56.83±0.60 | 40.44± 0.74 | 33.88± 1.08 | 72.73±0.32 | 56.43±0.46 | 40.01± 0.81 | 34.05± 1.33 |
| L2 | 59.07±1.81 | | 52.73±1.05 | 40.41± 1.78 | 59.39±2.51 | 62.05±0.60 | 51.43±1.22 | 40.63± 1.72 |
| L3 | 33.18± 1.09 | 49.95± 0.72 | | 62.79±1.01 | 33.07± 0.77 | 50.20±1.05 | 62.57±0.22 | 64.12±0.67 |
| L4 | 29.70± 0.82 | 40.12± 0.39 | 58.87±0.39 | | 30.81± 0.84 | 41.31± 0.25 | 59.60±0.39 | 68.80±0.06 |
| L5 | 73.26± 0.63 | 57.04±0.91 | 41.14± 0.71 | 35.02± 0.44 | | 56.70±0.70 | 40.89± 0.64 | 35.10± 0.38 |
| L6 | 57.60±3.65 | 60.62±1.78 | 51.98±1.25 | 41.79± 2.88 | 57.86±3.44 | | 51.52±1.79 | 42.38± 3.29 |
| L7 | 31.39± 1.29 | 48.18± 0.89 | 62.86±0.52 | 64.07±1.20 | 31.54± 1.94 | 48.46± 0.66 | | 65.36±1.07 |
| L8 | 30.99± 1.93 | 41.21± 0.65 | 59.09±0.68 | 69.12±0.48 | 32.09± 1.90 | 42.16± 0.29 | 59.84±0.35 | |

Environment: Skewed & non-IID

| | L1 | L2 | L3 | L4 | L5 | L6 | L7 | L8 |
|---|---|---|---|---|---|---|---|---|
| L1 | | 66.09±0.97 | 42.84± 0.86 | 29.77± 0.76 | 58.45±1.06 | 56.38±0.47 | 38.90± 0.52 | 28.07± 0.74 |
| L2 | 66.99±0.65 | | 51.77±0.48 | 33.51± 0.66 | 48.36± 0.85 | 54.60±0.42 | 43.30± 0.36 | 29.72± 0.63 |
| L3 | 42.59± 1.51 | 50.25±1.42 | | 66.38±3.14 | 35.20± 2.19 | 46.84± 0.99 | 65.29±1.99 | 59.76±4.53 |
| L4 | 36.79± 0.86 | 41.14± 0.73 | 62.60±1.41 | | 35.63± 1.22 | 41.70± 0.57 | 67.31±1.31 | 82.31±0.92 |
| L5 | 58.25±0.55 | 51.20±0.98 | 36.05± 0.74 | 32.79± 1.24 | | 57.65±0.41 | 35.81± 0.66 | 31.48± 0.96 |
| L6 | 63.94±1.38 | 57.05±1.74 | 37.25± 1.15 | 30.48± 2.87 | 70.76±2.71 | | 34.65± 1.40 | 29.03± 4.52 |
| L7 | 35.66± 1.29 | 41.30± 1.60 | 64.11±0.86 | 78.31±1.19 | 33.84± 1.30 | 43.11± 1.33 | | 80.62±2.31 |
| L8 | 39.60± 0.20 | 40.83± 0.17 | 55.53±0.79 | 77.06±0.72 | 39.73± 0.19 | 42.48± 0.11 | 63.63±0.33 | |

Table 7: Matrix of the membership inference attack accuracy on a per learner basis for the 2D-slice-mean model across every federated learning environment. Rows are the attacking learner and columns are the attacked learner. Colored cells indicate successful attacks and more heated cells specify higher attack accuracies. The results are over 5 random runs.

example, for the Uniform & non-IID distribution, learners L1 and L5 have a similar distribution and hence the attack from L1 on L5 or vice-versa has higher accuracies. However, the attack vulnerabilities are not symmetric; for example, the accuracy of the attack from L3 to L8, or L7 to L4 is higher than vice-versa, even though both learners have trained on the same number of samples. Such differences may be due to the neural network's tendency to overfit differently over diverse local data distributions, which in this case is the age range. An adversary with some more privileged information like knowledge of the distribution of labels or outputs will design more sophisticated attacks.

## Appendix C. Attack Architecture and Training Details

### C.1. Attack Classifier Parametrization

We train deep binary classifiers that take different features as input and output the probability of the sample being in the model's train set or not. We presented the importance of different features derived from a sample and the trained model for membership inference attacks in Section 4.1. In the case of a black-box attack, the attacker can only use the model's output. In contrast, in the case of white-box attacks, the attacker may also exploit the knowledge of the model's internal working. We have used gradient and activation information to simulate the attacks.

We repurpose the model's architecture to create a binary classifier for preliminary experiments on using activations for attacks. For example, in Figure 6(a), when simulating an attack that use activations from second hidden layer, i.e, after `conv 2` layer, we used a classifier that had layers from `conv 3` to `output`. However, as discussed in Section 4.1, the activations are not very useful features for membership attacks, and we did not do further experiments with activations.

To compute membership inference attacks using only the error feature, a 1D feature, we have used a random forest classifier. For other features, i.e., prediction, labels, gradients, and gradient magnitudes, we have used a generic setup where each feature is embedded to a 64-dimensional embedding using their respective encoders. The embeddings are then concatenated and passed through a dropout layer and a linear layer to output the logit. Below we describe the architecture of the encoder for each feature. We do not do an excessive architecture search but observed that the results are not very sensitive to the specific encoder architecture.

- **Prediction and label:** Prediction and label form a two-dimensional continuous feature. To create the embeddings, these are processed via a linear layer and `ReLU` non-linearity.

- **Gradient magnitudes:** We use parameter-gradient magnitudes of each layer as features resulting in a 14 dimensional feature for `3D-CNN` and an 18 dimensional feature vector for `2D-slice-mean` model. These are processed via a linear layer and `ReLU` non-linearity to generate the embedding.

- `conv 1` **gradients:** The size of `conv 1` gradient feature is 288 ($3 \times 3 \times 1 \times 32$) and 864 ($3 \times 3 \times 3 \times 1 \times 32$) for `2D-slice-mean` and `3D-CNN`. We project this feature vector to the desired embeddings size (64) by using a linear layer followed by `ReLU` non-linearity.

- **conv 6 gradients:** For `3D-CNN`, the feature dimension is $1 \times 1 \times 1 \times 256 \times 64$. We reshape it to $1 \times 256 \times 64$ and then process it through three convolutional blocks consisting of a 2D-convolution layer, max-pool and `ReLU` non-linearity with 64, 64, and 16 output filters. Finally, we pass the resulting activation of size $16 \times 6 \times 6$ through a linear layer and `ReLU` non-linearity to get the desired 64-dimensional embedding. The convolution kernel sizes were $5 \times 5$, $4 \times 2$, and $4 \times 3$ and the max-pool kernel sizes were $4 \times 2$, $4 \times 2$, and $2 \times 2$.

  For `2D-slice-mean`, the feature dimension is $1 \times 1 \times 256 \times 64$. We reshape it to $64 \times 256$ and process it through three convolutional blocks consisting of a 1D-convolution layer, max-pool, and `ReLU` non-linearity with 128 output filters in each layer. Finally, we process the resulting activations of size $128 \times 14$ through a linear layer to get the embedding. The convolution kernel sizes were 5, 4, and 3. The 1D-max-pool kernel sizes were 4, 2, and 2.

- **output gradients:** This layer has different number of parameters for both the models, and so we used different encoders. For `2D-slice-mean` model, two final feed-forward layers are considered as `output` layers.

  For `3D-CNN`, the feature dimension is $1 \times 1 \times 1 \times 64 \times 1$. It is reshaped to a 64-dimensional vector and passed through the linear layer and `ReLU` non-linearity to get the embedding.

  For `2D-slice-mean`, we consider two final feed-forward layers as the `output` layer, one of the layers has dimensions $64 \times 1$ and is encoded similar to `3D-CNN`'s `output` layer. The other feed-forward layer parameters are $32 \times 64$. We process it through three convolutional blocks consisting of a 1D-convolution layer, max-pool, and `ReLU` non-linearity with 64 output filters in each layer. Finally, we process the resulting activation of size $64 \times 4$ through a linear layer to get the 64-dimensional embedding. All the convolution kernel sizes were set to 3 and 1D-max-pool kernel sizes were 2.

**Note:** When using features from multiple trained models for attack (e.g., in case of federated training), we compute the logits using the deep classifiers described above and use the average logit to compute the probability. The classifier parameters are shared across features from different models. The main intuition to consider averaging is that averaging in log space would mean considering prediction from each trained model's feature independently.

## C.2. Training

To train attack models, we used `Adam` optimizer, a batch size of 64 and $1e^{-3}$ learning rate. We trained the models for a maximum of 100 epochs with a patience of 20 epochs, and chose the best model by performance on the validation set, created by an $80 - 20$ split of the training data. Training data creation for the attack models is described in Section 3.2.

## Appendix D. Differential Privacy

Differential privacy was initially proposed as a mathematical framework to secure information about individual records while releasing group or aggregate query results on a database.

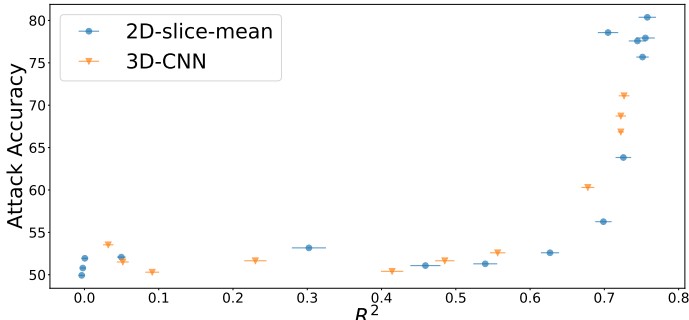

Figure 7: Attack Accuracy vs. $R^2$ for models trained with differential privacy. Error bars are generated by bootstrapping the test set 5 times using 1000 samples.

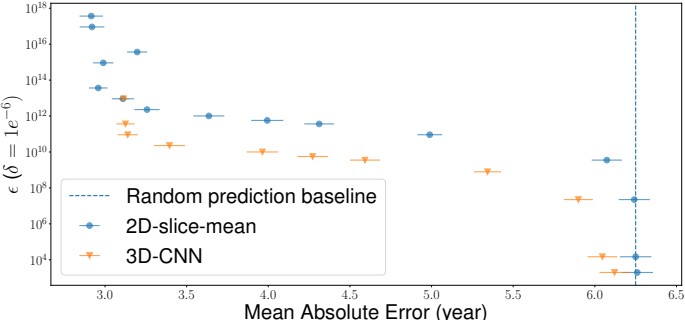

Figure 8: Performance (MAE) vs. $\epsilon$ at $\delta = 1e^{-6}$ for models trained with differential privacy. Differential privacy is a very strong notion of privacy. It destroys the performance to achieve any non-vacuous privacy guarantees. Error bars are generated by bootstrapping the test set 5 times using 1000 samples.

It is adapted to machine learning by considering model parameters as the output and training dataset as the database. Differential private machine learning aims to learn a parametric model so that the final parameters do not differ much if trained on another dataset differing from the original dataset by a single sample. The privacy parameter $\epsilon$ quantifies the difference, and lower $\epsilon$ is more private (Dwork and Roth, 2014; Abadi et al., 2016). More formally, a randomized algorithm $\mathcal{A} : \mathcal{D} \rightarrow \mathcal{W}$ is $(\epsilon, \delta)$-differential private, if for any two adjacent inputs $d, d' \in \mathcal{D}$.

$$Pr[\mathcal{A}(d) \in w] \leq e^{\epsilon} Pr[\mathcal{A}(d') \in w] + \delta \quad \forall \, w \in \mathcal{W}$$

For non-vacuous guarantees, $\epsilon$ is desired to be lower, usually less than 1. However, there is no standard agreement on how much is sufficiently small (Laud and Pankova, 2019). $\delta$ depends on the dataset size and is desired to be less than $N^{-1}$, where N is the dataset size. In the specific case of supervised deep learning, output is the neural network parameters, input is the labeled dataset, and algorithm is the training algorithm (usually some variant of SGD). Intuitively, to ensure strong privacy guarantees, it is desired to minimize a single training sample's influence on the neural network parameters.

We have used differential private version of SGD (DP-SGD) proposed by Abadi et al. (2016) which achieves privacy guarantees by adding Gaussian random noise to the gradients of each sample and implemented in `pytorch-opacus`[5]. This procedure avoids learning too much information about a single sample, thus providing privacy guarantees. In practice, we have used the `Adam` variant of DP-SGD with a learning rate of $5e^{-5}$, emulating the same training setup as Gupta et al. (2021).

Differential privacy assumes a powerful and worst-case adversary, which may be unrealistic. We find that to achieve non-vacuous privacy guarantees ($\epsilon < 100$) with differential privacy amounted to losing the performance altogether on the brain age prediction problem (see Figure 8). However, even with vacuous guarantees, we see that differential privacy could reduce the vulnerability to realistic membership inference attacks as shown in Figure 3(a) and Figure 7.

## Appendix E. Membership Inference attacks in centralized setup without the knowledge of training samples

In the setup of Section 4.1, we assumed that the adversary has access to some training samples to perform membership inference attacks. However, such an assumption may be too restrictive. Here, we discuss white-box membership inference attacks and the attacker has access to some samples from the training distribution only instead of training samples. The attacker does not know if these samples were part of training.

Since the attacker does not have access to the samples used to train the model, attack classifier cannot be trained. To circumvent this limitation, we use the idea of shadow training from Nasr et al. (2018). Briefly, the attacker trains new models with the same architecture and training hyperparameters using the samples available from the training distribution. These newly trained models, called shadow models, are expected to imitate or shadow the trained model's behavior — for example, similar overfitting behavior, similar training performance, etc. Therefore, the attacker may train the attack classifier using the shadow models and samples used to train them and expect to transfer to the trained models he intends to attack.

### E.1. Setup

For this section, we use the train, test and validation split described in Section A.1. We consider that the attacker has access to a trained model and some samples from the training distribution, which may or may not overlap with the samples used to train the model being attacked. The attacker intends to identify if some data sample was used to train the model.

The attacker is trying to attack the same models that are described in Appendix A. These models are trained on the full training set. To simulate the attacks with access to only the training distribution but not training samples, we consider the scenario where the attacker has access to 5000 random samples from the training distribution. For this, we pick 5000 random samples from the original training set of size 7312. The attacker is trying to determine the membership of samples from the train set, which differ from these 5000 samples. Due to limited data, the data used to train the shadow models overlaps with the

---

5. https://github.com/pytorch/opacus

| Model | Test | Validation |
|---|---|---|
| 3D-CNN | $71.74 \pm 1.82$ | $75.22 \pm 0.22$ |
| 2D-slice-mean | $74.39 \pm 2.14$ | $85.46 \pm 0.24$ |

Table 8: Membership inference attacks without the knowledge of training samples. The test performance results from performing membership inference on the trained model using attack models trained on information from the shadow model. The validation performance is the attack classifier's performance on the validation set derived from the shadow models' training set.

data used to train the original model. A more difficult scenario will be if these datasets do not overlap at all.

### E.2. Result

To report the membership inference attack performance, we created a test dataset of 1500 samples from the full train set (different from 5000 samples that the attacker already has) and 1500 samples from the unseen set to evaluate the membership inference attack accuracy. We trained a single shadow model with 5000 samples that are available to the attacker. The attack classifier is trained to attack the shadow models similar to earlier experiments using prediction, label, and gradient of `conv6` and `output` layers from the shadow model as the features. We extract these features from the trained model and classify them with the attack classifier to infer the memberships. The results are summarized in Table 8. The 'Test' column shows the result of performing a membership inference attack on the trained model, which is what we are interested in. We also report the attack accuracies on the validation set derived from the shadow model's training set in the 'Validation' column. We observe that even without access to training samples, the membership inference attacks are feasible, albeit with slightly lower accuracy than the case in which the adversary has access to some of the training samples.

