# OpenReview forum: "Membership Inference Attacks on Deep Regression Models for Neuroimaging"
_MIDL.io/2021/Conference — MIDL 2021_

### Official Review · ~Christopher_P_Bridge1 · 2021-03-07

**Confidence:** 4
**Preliminary Rating:** 4
**Recommendation:** Oral
**Final Rating:** 4

**Summary:**

This paper investigates the effectiveness of membership attacks on medical image analysis models in both centralised and federated training scenarios. The chosen application is brain age prediction from brain MRI. The results demonstrate that membership of the training set can be determined with an accuracy of over 80% in certain situations, and that membership attacks are more accurate on average in the centralised setting.

**Strengths:**

This paper is distinguished by the clarity of its communication, the thoroughness of its references to related literature, and the  breadth of the experiments performed.

Since federated learning is beginning to become more common in medical image analysis settings due to privacy concerns, this is also a timely paper and important for the community and regulatory bodies in order to be able to assess the potential consequences of data sharing.

Overall this is an excellent submission.


**Weaknesses:**

In my opinion the major weakness of the paper is that the scenario studied in this paper is somewhat unrealistic. Having access to the "white box" model is clearly realistic in a federated learning setting but in order to infer the membership of a certain individual in the training set the attacker must have access to the individual's brain MRI. Since this in itself would already be a major breach of privacy, the ability to determine membership of a certain training is unlikely to be of particular further concern. Howeever, I believe it is still for the community to understand the implications of data sharing.


**Deanonymize Review:**

yes

**Detailed Comments:**

Since a large part of this paper relates to federated learning, and this is especially relevant to the MIDL community, I would consider including a reference to FL in the title.

Referring to the number of learner combinations where the attack accuracy over all samples was > 50% as simply "the number of successful attacks" is a little confusing. This sounds like it could mean the number of individual images whose membership in the training set was revealed.

Although the details may be available in the references, the authors should give some further details about the MRIs used in the experiment. What type (contrast) scan was used for example? Were they all taken from patients with no major abnormalities?


**Final Rating Justification:**

I feel the authors have done  a good job clarifying questions from myself and other reviewers and so have not changed my recommendation of accept. I still believe this is an interesting and important paper to include in the conference.

I take their point in response to my question about the premise, but I still feel that the situation in which a lab both has access to brain MR *and* would have anything further to learn from a membership attack seems relatively unlikely.

**Justification Of The Preliminary Rating:**

This is a very clearly presented set of thorough experiments with results that will be of interest to the medical image community, especially those concerned about the privacy implications of sharing deep learning models and training data.

**Paper Type:**

validation/application paper

**Questions To Address In The Rebuttal:**

It is not clear which of the two experimental settings (centralised or distributed) the results in Figure 3 relate to.

Furthermore, what are the different models that make up the datapoints in figure 3b and what distinguishes them from each other? Was the extent of overfitting deliberately controlled or did it arise naturally? Please clarify in the text

In this work the authors have decided to study a regression problem. Do they believe that these results would generalise to perhaps more common medical image analysis problems such as classification and segmentation?

**Special Issue:**

yes

---

> ### Author Response · Authors · 2021-03-18
> **Response to weaknesses and detailed comments**
>
> We thank the reviewer for highlighting the relevance of our paper for the medical imaging community and for an encouraging review. Below we have addressed their concerns.
>
> === Response to weaknesses ===
>
> We want to clarify the reviewer's concern about having an unrealistic setup of having access to an individual's brain MRI. We want to point out that in many cases, data is shared with different research groups. For instance, the UK Biobank dataset, which is the dataset we used, has approved 10,000 data access applications. These 10,000 labs might have more than 100,000 lab members in total who have access to personal health records (assuming on average ten people are in a lab using the data). Not all these applications analyze MRI, but they are all analyzing personal medical data, with in-depth information on people's personality, IQ, health conditions, addictions, mental health, etc. So having access to a medical research dataset is not unusual. Even if only 1 in 100,000 people use the data in an unauthorized way or a data breach happens in any of these labs, privacy could be broken.
>
> Moreover, even though data is not public, different groups are given access to different attributes. These attributes independently may not be revealing anything, but when seen together / connected via these attacks, they can reveal more information than intended. Lastly, our attacks in the centralized case can be seen as the worst-case scenario of the attacks.
>
> === Response to Detailed Comments ===
>
> - We have modified the phrase "the number of successful attacks" as "the number of successful instances of learner-attacker pairs" in the draft. We hope that this will improve the readability and avoid misinterpretations.
>
> - Due to space constraints, we had omitted the details about MRI preprocessing. The draft has been updated to reflect that we have used scans of healthy subjects with no abnormalities, and T1 structural MRI scans are used. We have used the same MRI preprocessing steps as [1, sec 2.2].
>
> [1] Lam, Pradeep K., et al. "Accurate brain age prediction using recurrent slice-based networks." 16th International Symposium on Medical Information Processing and Analysis. Vol. 11583. International Society for Optics and Photonics, 2020.

---

> ### Author Response · Authors · 2021-03-18
> **Response to Questions**
>
> > It is not clear which of the two experimental settings (centralised or distributed) the results in Figure 3 relate to.
>
> Fig. 3 studies the performance of differential privacy in the centralized setup. We have clarified this in the draft.
>
>
> > Furthermore, what are the different models that make up the datapoints in figure 3b and what distinguishes them from each other? Was the extent of overfitting deliberately controlled or did it arise naturally? Please clarify in the text
>
> We have trained models with Gaussian noise of different variance added to the gradients to achieve differential privacy. More noise makes the models more private but at the cost of performance (MAE). Fig. 3a shows membership inference attacks vs. MAE for models trained with different privacy budgets. Fig. 3b shows membership inference attacks vs. overfitting for these models. We did not control for overfitting explicitly, and it arose due to training with different privacy budgets or equivalently different gradient noise. We have clarified this in the new draft.
>
>
> > In this work the authors have decided to study a regression problem. Do they believe that these results would generalise to perhaps more common medical image analysis problems such as classification and segmentation?
>
> Any guaranteed claims regarding the generalization of attacks to other tasks (e.g., classification, segmentation, etc.) will require further empirical evaluations. Nevertheless, we believe that these results should generalize to other tasks due to the following reason:
>
> 1. Neural Networks are trained with gradient-based learning until convergence. Therefore, gradients of training samples are lower or similar in magnitude than unseen samples (for example, see Fig. 1). This property is characteristic of all gradient-based learning. So we believe these attacks should generalize to other tasks, too, where models are trained via backpropagation.
> 2. Many previous works like [2,3] have shown membership inference attacks for classification tasks but not medical imaging data.
> 3. We have shown that attack accuracies correlate with overfitting (Fig. 3b). Overfitting is a phenomenon common to most learning problems. Thus, these attacks should generalize to other learning tasks too.
>
> [2] Reza Shokri, Marco Stronati, Congzheng Song, and Vitaly Shmatikov. Membership Inference Attacks Against Machine Learning Models. In 2017 IEEE Symposium on Security and Privacy (SP), pages 3–18, 2017.
>
> [3] M. Nasr, R. Shokri, and A. Houmansadr. Comprehensive Privacy Analysis of Deep Learning: Passive and Active White-box Inference Attacks against Centralized and Federated Learning. In 2019 IEEE Symposium on Security and Privacy (SP), pages 739–753, 2019.

---

### Official Review · AnonReviewer4 · 2021-03-08

**Confidence:** 2
**Preliminary Rating:** 3
**Recommendation:** Oral, Poster
**Final Rating:** 3

**Summary:**

The authors demonstrate membership inference attacks, i.e. determining from a trained model whether a particular example was used in the training process, on a Brain Age prediction from MRI scans. They simulate attacks in both the centralised and distributed (federated) learning setup to demonstrate the vulnerability of these models. Their experiments suggest a link between model overfitting and their vulnerability to attacks, as quantified by a correlation measure.

**Strengths:**

1. The paper is very well written and structured, making it easy to read and follow, with definitions and the problem setup explained carefully. In my opinion, this is a plus for readers that are new to federated learning and privacy in the context of medical imaging.

2. The authors evaluate both centralised as well as the federated learning setup on two different models, with trends in attack performance replicated in the experiments as presented.

3. The authors provide comprehensive evaluation of two main attack paradigms namely, the white box setup assuming the attacker has liberal access to the training setup and the model, and the black box setup which assumes limited access to model outputs. Their experiments support their conclusions.

**Weaknesses:**

One of the concerns with the results as presented is the lack of standard deviations/statistical comparisons provided for the performance in the federated setup (Fig. 2, Table 2, Figs 3 a and b, Figs 4 and 5). This would help indicate how robust/comparable the performances of different attackers are, and to what extent the trends with overfitting of the model vs vulnerability change with subsampling.

**Deanonymize Review:**

no

**Detailed Comments:**

Suggestions:

1. It would be good to also include the standard deviations of the mentioned accuracies over repeated experiments (for different dataset splits) in the federated setup. Mainly, this would help quantify the robustness of the attack performance in each case. For example, this would help solidify the performance trends in Tables 4 and 5

2. In a similar comment, it would be good if standard deviations over the MAE be reported for Figs. 3 a and b. Additionally, including a second metric for explained variance, such as the R^2 could also be useful, since in theory a better MAE could imply better estimation of the mean.



**Final Rating Justification:**

The authors have clarified the points I raised initially. I understand that re-running all the models with various splits may be prohibitive currently, but might be useful in the context of future extensions of this work.

I appreciate the clarifications to my detailed comments and I stand by my original decision to recommend acceptance.

**Justification Of The Preliminary Rating:**

Overall, I think the paper is well written, with the experiments convincing and comprehensive enough and clearly presented for a methodological paper in MIDL. There are a few points as mentioned above which can be better addressed, possibly in the rebuttal. The paper in my opinion is above the acceptance threshold.

**Paper Type:**

methodological development

**Questions To Address In The Rebuttal:**

Along with the comments above, could the authors please clarify how the training parameters (epochs, learning rate etc) for the federated setup (A.3) were determined, since the authors mention that no validation set was used?

**Special Issue:**

no

---

> ### Author Response · Authors · 2021-03-18
> **Response**
>
> We thank the reviewer for highlighting the strengths of our paper. We remark that it is computationally prohibitive to run full brain age training on newer dataset splits. We have tried our best to address the reviewer's concerns below.
>
> **Tables 2, 3, 4, and 5**
>
> Due to time and compute constraints, we could only report accuracies and standard deviation of multiple attacks over the already trained federation models. We have updated Tables 2, 3, 4, and 5 accordingly. We did not have the computational capacity to investigate the attacks' performance over federated models trained on different data splits. However, we emphasize that the three distributions we investigate in this work capture a large spectrum of challenging federated learning environments, especially the Skewed and non-IID, and stress the effectiveness and the vulnerability of the community models.
>
> **Fig 3a, 3b**
>
> Each point on the trend curve is a different brain age model trained from scratch. It is computationally prohibitive to train brain age models with newer dataset splits within the rebuttal timeline. However, we emphasize that each model on the trend curve is trained and attacked with unknown but different initialization of the neural network. Thus, the trend curves already encode some randomness. To have some idea about MAE variation, we have reported the errors by bootstrapping the test set.
>
> We thank the reviewer for the suggestion about the R^2 metric. We have added the results with R^2 metrics in the Appendix (Fig. 7).
>
> === Response to Questions ===
>
> Similar to the original federated learning work[1], our federated training setup used all the available local data samples for training. The selected hyperparameters were based on the performance of the centralized model against its validation set. In particular, for the learning rate and batch size, we originally performed a grid search over different combinations of hyperparameter values. We observed that we reached the best possible performance for batch size 1 and a learning rate equal to 0.00005. Henceforth, we employed these values for all the federated learning environments and kept them constant throughout execution. In terms of communication frequency (i.e., number of local epochs per learner), once we identified the best values for batch size and learning rate, we performed another grid search (against the test set in this case) for the Uniform & IID learning environment over the following set of total local epochs per learner: {1, 2, 4, 8, 16} and we picked number 4 as the number of epochs since it exhibited the best test performance for the community model. For the non-IID environments, we used the same hyperparameters as in the IID case.
>
> [1] McMahan, Brendan, et al. "Communication-efficient learning of deep networks from decentralized data." Artificial Intelligence and Statistics. PMLR, 2017.

---

### Official Review · AnonReviewer3 · 2021-03-08

**Confidence:** 2
**Preliminary Rating:** 2
**Final Rating:** 3

**Summary:**

The paper investigates the effect of membership inference attacks on brain age regression networks in the setting of regular training and training with federated learning. The authors compare different types of access to the network in terms of activations, gradients or outputs. Various experiments show that MIA attacks are feasible with accuracies ranging from `~60-80%.

**Strengths:**

The paper tackles an important problem of whether one can discern whether someone's data was used in a training set. The paper considers the setting of centralised training as well as federated learning and tests different networks and available features to discern the training set membership. The authors also consider different underlying data distributions to test the robustness of MIA attacks to non uniform distributions.

**Weaknesses:**

The paper has little methodological novelty but instead explores the usage of MIA attacks on the medical domain. However, especially here it would be important to elaborate on different data access scenarios and how an attacker would obtain access to the data or whether they contributed data themselves. Further, the paper would benefit from further discussion of the impact of the possibility of those attacks and the specific attack performances. Lastly, the attacks assume access to training data in the centralised setting - this is a very strong assumption and one should explore the usage of samples from the training distribution instead. However, here the non-members come from the same distribution as the training samples. This should be discussed and would hugely improve the setting of MIA attacks in the medical context.

**Deanonymize Review:**

no

**Detailed Comments:**

- Why is BrainAGE capitalised the way it is? Brain age regression is a fairly common problem.
- How big is the overall training data?
- What are the validation errors of the different models?
- What does it mean that results are averaged over multiple attacks? Are those multiple attack models?
- How are the layers chosen that are used for the attack model?
- Do you have any intuition why activation based attacks don't work that well?
- How useful is an attack with 60% accuracy really? How can an attacker tell whether their attack would be successful in this setting? Is the validation loss of the attacker a reliable indication for it's performance?
- It'd be useful to expand the description of table 2 and plain the meaning of the different distribution types there.
- How is overfitting in Fig 3.b) measured?
- On which setting were the experiments with differential privacy performed?


**Final Rating Justification:**

The authors sufficiently addressed my comments to push my decision over the boundary - I believe this paper is still borderline but happy to vote for a weak accept. While the paper provides little methodological novelty, it brings attention to an important area of research for the medical imaging community.

**Justification Of The Preliminary Rating:**

I would have liked to score this as a borderline paper because of the lack of novelty and lack of justification for some of the data access assumptions to the training and non-training data. In the domain of medical application one should thoroughly justify those assumptions with relevant attack scenarios and consider which cases could actually happen in the real world. However, I am happy to reconsider my rating and confess little experience with differential privacy and MIA attacks.

**Paper Type:**

validation/application paper

**Special Issue:**

no

---

> ### Author Response · Authors · 2021-03-18
> **Response to weaknesses**
>
> We thank the reviewer for a critical and thorough review. Below we address their concerns.
>
> === Response to Weaknesses ===
>
> We thank the reviewer for their valuable comments on improving the paper. We acknowledge their remark on elaborating more on how the different data access scenarios for an attacker can arise in the learning domains we investigated. In the current setting, we posit that an attacker can obtain access to the data records by some data leak, public dataset, or data repurposing (centralized setup), or a participating site can itself act as malicious (federation setup).
>
> Attacks in the centralized setup give us an idea of the worst-case scenario concerning privacy and are also used to evaluate the performance of different features. We thank the reviewer for inspiring us to investigate attacks using only the training distribution samples instead. For such attacks, we use the idea of shadow training from [1]. With these attacks, assumptions about having access to training set samples can be removed. Instead, the attacker trains a new brain age model using public data drawn from the same distribution as the brain age model trained on private data. This model is used to train a membership attack that can then be applied to the model trained on private data. Thus, the attacker only requires a knowledge of training distribution rather than training samples. We have added results with the shadow training in the Appendix.
>
> [1] Reza Shokri, Marco Stronati, Congzheng Song, and Vitaly Shmatikov.  Membership Inference Attacks Against Machine Learning Models.  In2017 IEEE Symposium on Security and Privacy (SP), pages 3–18, 2017

---

> ### Author Response · Authors · 2021-03-18
> **Response to detailed comments**
>
> > Why is BrainAGE capitalised the way it is? Brain age regression is a fairly common problem.
>
> BrainAGE was used to emphasize Brain Age Gap Estimation (BrainAGE), clarified in Sec. 2.1. We have updated the draft to avoid any confusion regarding this.
>
> > How big is the overall training data?
>
> The size of the training dataset for both the setups is described in A.1 and A.2. The size of samples used for training attack classifiers is described in sec 3.2. In particular, for centralized setup training/test/validation set sizes were 7312/2194/940. For the federated learning setup, training/test sizes were 8356/2090.
>
> > What are the validation errors of the different models?
>
> For centralized training, test/train/validation MAE is:
>
> |   |Train   | Test   |Valid   |   |
> |---|---|---|---|---|
> | 2D-slice-mean | 0.77 | 2.88 | 2.92 |
> | 3D-CNN | 1.39 | 3.13 | 3.09 |
>
> For federated training, train/test MAE is:
>
>
> |      | Train / Test  (Uniform IID) | Train / Test  (Uniform Non-IID) | Train / Test (Skewed Non-IID) | |
> |---|---|---|---|---|
> | 3D-CNN | 2.16 / 3.01 | 3.41 / 3.81 | 2.83 / 3.47 |
> | 2D-slice-mean | 1.81 / 2.76 | 2.40 / 2.98 | 2.42 / 3.10 |
>
> Similar to the original federated learning work [2], the federated learning setup did not use a validation set. We have updated the draft with these tables (Tables 6 & 7).
>
> > What does it mean that results are averaged over multiple attacks? Are those multiple attack models?
>
> We considered all possible attacker-learner pairs (56) and averaged the results. Each learner may act as an attacker and attack seven other learners, leading to 8x7=56 attacks. A separate model is trained for each attacker. Thus, these are multiple attacks --- attacker attacking each learner and multiple models --- each learner may act as the attacker. Due to space constraints, we provide more detailed results in Appendix B.
>
> > How are the layers chosen that are used for the attack model?
>
> Table 1 shows the results of using different features for attacks. The rationale behind using these features is partly explained in footnote 3, page 6. For the federated learning setup, we have used prediction + label + gradient in Table 2 (see sec 4.2, para 2, line 5). Attacks with other features are discussed in Table 3 and Appendix B.
>
> > Do you have any intuition why activation based attacks don't work that well?
>
> Train set and unseen set are sampled from the same distribution. Therefore, the activations of both sets are expected to have a similar distribution and, as a result, hard to distinguish. On the other hand, neural networks are trained till convergence, i.e., till the gradient of parameters w.r.t loss becomes zeros. Therefore, gradients of training samples are similar or often lower in magnitude than unseen samples. Thus, the gradients work better than activations for these attacks.
>
> > How useful is an attack with 60% accuracy really? How can an attacker tell whether their attack would be successful in this setting? Is the validation loss of the attacker a reliable indication for it's performance?
>
> We acknowledge the reviewer's criticism about the attacks being only 60% accurate in the case of federated learning. Even though an attack with 60% accuracy may not be very useful, we would like to point the reviewer to Figures 2 and 3b in particular. In Fig. 2, we show that the attack accuracy increases with the federation rounds. Fig. 3b sheds light on this phenomenon by showing the relation between overfitting and attack accuracy. As models are trained for more iterations, they tend to overfit, and their vulnerability increases. We think this evidence about overfitting and increasing accuracy with federation rounds warrants concern about privacy.
>
> Moreover, the datasets we have used are some of the bigger datasets in neuroimaging. Smaller datasets are expected to have more overfitting and hence may be more vulnerable.
>
> We have not explored the validation loss of the attacker in the federated setting. Therefore we cannot safely claim that it is a reliable indicator for the attacker's performance.
>
> > It'd be useful to expand the description of table 2 and plain the meaning of the different distribution types there.
>
> The details about data distribution are provided briefly in Sec 3.2 and in more detail in Appendix A.2. We have also refined the caption of Table 2 in the new draft.
>
> > How is overfitting in Fig 3.b) measured?
>
> Overfitting is measured as the difference between test and train performance (MAE). We have indicated this in sec 4.3, last paragraph, third last lime.
>
> > On which setting were the experiments with differential privacy performed?
>
> The differential privacy experiments were performed on the centralized setup. We have updated this information in the draft.
>
> [2] McMahan, Brendan, et al. "Communication-efficient learning of deep networks from decentralized data." Artificial Intelligence and Statistics. PMLR, 2017.

---

### Official Review · AnonReviewer2 · 2021-03-08

**Confidence:** 4
**Preliminary Rating:** 2
**Final Rating:** 3

**Summary:**

The paper presents an analysis of membership inference attacks: a setup whereby an attacker can ascertain whether a sample was used during model training or not. This knowledge should be able to be inferred from the model as the privacy of individuals would be compromised. This is of particular importance in clinical settings.

In the setting devised by the authors, membership can be predicted with accuracies ranging from 56 % to 83 % depending on features used.

**Strengths:**

Privacy is vitally important in clinical settings and individuals who partake in medical studies or databases should reasonably expect that third parties should not be able to infer whether they took part in a study or not.

The motivation behind this paper is clear: demonstrate that this anonymity is broken under the assumptions set out in the experiments.

The paper clearly discussed the relevant literature and background domains relevant for this interdisciplinary work. Most of it is well written.

**Weaknesses:**

The major weakness of this paper is the lack of clear assumptions set out for the experiments, particularly with respect to the distributions of the training and unseen sets.

In 3.2 the attack setup is described. It states that the attacker has access to some of the training samples and some samples that were not used for training. However, it is not stated whether the samples in these two sets come from the same distribution and are sampled i.i.d. Surely that is something that the attacker must know in order to train a classifier.

Furthermore, the claims in the abstract and introduction that "However, we demonstrate that allowing access to parameters may leak private information, even if data is never directly shared." and  "In particular, we show that it is possible to infer if a sample was part of the training set that was used to train the model **given only access to the model prediction (black-box) or access to the model itself (white-box).}**". However, the actual attack uses 1500 samples from the training data.


**Deanonymize Review:**

no

**Detailed Comments:**

- In the fourth sentence of the results, there is a statement saying that the gradient magnitudes computed from the rained model should be useful for to distinguish samples used during training to those that aren't. This isn't obvious and some extra details would be useful here.

- All tables and figures are missing units for error

- Use of UK Biobank data must be acknowledged as: “This research has been conducted using the UK Biobank Resource under Application Number xxxxxx” see: https://www.ukbiobank.ac.uk/media/vo2p3ezn/access_017-faqs-v3-3-1.pdf

**Final Rating Justification:**

The authors have responded to my questions and made changes to the manuscript that address my concerns. I revise my rating for this paper to a 3 (weak accept). If available, I would have chosen a 3.5 (accept).

**Justification Of The Preliminary Rating:**

The paper presents an important aspect of privacy in medical imaging settings: membership inference attacks. The paper is well written and for the most part clearly sets out and demonstrates this drawback. My concern regarding the assumptions required for the attacks to be possible in practice, inconsistencies surrounding the claims in abstract/intro and acknowledgement of UK Biobank data are the only thing holding this paper back from an accept.

**Paper Type:**

validation/application paper

**Questions To Address In The Rebuttal:**

- Clarifications surrounding the limitations surrounding source distributions of the data:
     - Must the sample being attacked come from the same distribution as the leaked training data?
     - Is there any limitation on the relationship between the training and unseen data used for the classifier?

- The claims in the abstract and introduction should be corrected to match the actual setup

**Special Issue:**

no

---

> ### Author Response · Authors · 2021-03-18
> **Response to weaknesses and detailed comments**
>
> We thank the reviewer for their critical review. Below we have addressed their main concerns.
>
> === Response to Weaknesses ===
>
> **Distribution of training and unseen set**
> - For the centralized learning setup, the distribution of the training set and unseen set for training the classifier of the attacker are IID samples from the same distribution. We have updated footnote 1 to reflect this.
>
> - For the federated learning setup, when training the attack model, the unseen set is the same as the testing set, and the train set is the attacker's train set. Unlike centralized setup, the distribution of the unseen set and training set that the attacker model is trained on could be different, particularly in non-IID environments. We have clarified this in Sec 3.2.
>
> **Direct access to the training data**
> - In the centralized setup, we assume access to random samples from the training set. These attacks give us an idea of the worst-case scenario concerning privacy and the effectiveness of different features. However, these assumptions about having access to training set samples can be removed by using the idea of shadow training [1], where the attacker will re-train the model on the new training data sampled IID from the training distribution and train the attack model on the gradients from this newly trained model. We have added results with the shadow training in Appendix E.
>
> - Attacks in the federated setup do not have "direct access" to other learners' data samples. That is, they do not train the model on other learner's training data.
>
> We thank the reviewer for highlighting the mismatched claims. We have fixed the abstract and the introduction in the updated draft. However, with the shadow training results, we think that not having “direct access” is justified. The access to samples required for making these attacks can be indirect, i.e., through some data leak, public dataset, or data repurposing.
>
> === Response to Detailed Comments ===
>
> Neural Networks are trained with gradient-based learning until convergence. The convergence is achieved when the gradient of loss w.r.t parameters on the training set is close to 0. Therefore, gradients of training samples are expected to be lower in magnitude than unseen samples. This property is characteristic of all gradient-based learning. We have updated the draft to reflect this.
>
> We thank the reviewer for pointing out the missing units and UK Biobank acknowledgment. We have fixed the figures and tables and also added error bars wherever appropriate.  UK Biobank is acknowledged in the updated draft.
>
> [1] Reza Shokri, Marco Stronati, Congzheng Song, and Vitaly Shmatikov.  Membership Inference Attacks Against Machine Learning Models.  In2017 IEEE Symposium on Security and Privacy (SP), pages 3–18, 2017

---

> ### Author Response · Authors · 2021-03-18
> **Response to Questions**
>
> > Must the sample being attacked come from the same distribution as the leaked training data?
>
> In the federated setup, we see the evidence that the attacks are more successful when the sample being attacked is from the same distribution as the training set samples used to train the classifier. As it is apparent from Tables 4 and 5, the distribution of the training set samples on which the attack models are trained can improve or deteriorate the attack performance. In particular, for the uniform and IID cases, we see that in all the scenarios, the attack is successful (>50%) since learners' local training dataset follows a similar distribution. However, as we move towards harder learning environments, where the local training datasets have highly heterogeneous data distributions (Uniform & non-IID, Skewed & non-IID), the success rate of the attacks degrade, and the learners with similar local data distribution to that of the attacker are successfully attacked.
>
> For instance, in Table 4 of Appendix B, for the Skewed & non-IID environment, if we consider Learner 6 as the attacking learner and learner 5 as the attacked learner, the accuracy is equal to 70.74%. This phenomenon can be attributed to the similar local training distributions, shown in Figure 4(f) of Appendix A; for learner 6 the mean brain age of local MRI scans is equal to 65.91 and for learner 5 equal to 70.82.
>
> > Is there any limitation on the relationship between the training and unseen data used for the classifier?
>
> The training and unseen set are expected to be from the same distribution. Otherwise, one may train a classifier just using raw data samples as the features directly without using the gradient features. In the case of federated learning experiments, the global distribution of the training set, i.e., if all data were pooled from across the learners, is the same as the unseen set. However, to train the classifiers, due to the data access restrictions, we only used the individual learner's train set (as training samples) and the global test set (as the unseen samples). Therefore, the attack classifier may learn features that are not useful to identify characteristics of the global train set. For example, the classifier trained with training samples from a specific learner with only young age subjects may learn to identify older age groups as the test set always, which is not true for the global train set. This behavior of the classifier may be because the attacking learner may not have that age group in its training set.
>
> Forcing the distribution of train set samples and unseen set for training the attack classifier to be the same may improve the performance by learning more specific discriminative features, but these would need further empirical investigation.

---

### Author Response · Authors · 2021-03-18
**Summary of main changes to the draft**

We thank all the reviewers for their valuable feedback. We have made updates to the draft to address reviewer comments. To demonstrate the more realistic attacks with less information access, we added new results for attacks in the centralized training setup that require no examples from private training data. However, we also emphasize that in large neuroimaging collaborations, some data leakage is likely, as we highlighted in response to reviewer #1. Our attacks demonstrate the feasibility of amplifying a leak from one party in a manner that endangers privacy in the whole consortium. We have highlighted major changes with purple color for convenience. Below is the summary of major changes we made to the manuscript:

**Main Paper**
- We have updated the abstract claims to be consistent with the experiments as per reviewer #2’s feedback.
- In Section 3, we have added details about MRI preprocessing as per reviewer #1’s feedback and added discussions about data distribution to address reviewer #2’s comments.
- In Section 4.1, we have clarified why gradients as features are useful for membership inference attacks to address reviewer #2‘s feedback.
- We have updated Section 4.2 to clarify some of the presented federated learning terminologies and results.
- Tables 2, 3, 4, and 5 are updated with results from multiple attack-runs as per reviewer #4’s suggestions.
- Fig 3a, 3b are updated with MAE standard deviations by bootstrapping the test set as per reviewer #4’s suggestions.
- We updated the acknowledgments section to include the UK Biobank Resource.

**Appendix**
- We have highlighted the brain age model performances in Tables 6 and 7 as per reviewer #3’s feedback.
- Inspired by reviewer #3’s comments, we have added new results for attacks in the centralized setup without the knowledge of private training samples in Appendix E.

---

### Meta-Review · Area_Chairs · 2021-03-29

**Recommendation:** Accept (Poster)

**Metareview:**

The paper discusses membership inference attacks in the specific application of (age) regression from neuro MR, which all reviewers agree is a timely and important topic.

The reviewers had several consistent comments and concerns that I feel are appropriate, and all felt that the comments from the authors helped improve their opinion of the paper (more below). Overall, I believe the paper is borderline leaning on accept (3 weak accepts and one strong accept after rebuttal), with most reviewers emphasizing this isn't their area of expertiese -- but of course, with a new topic, this is likely the case for much of the community.

The choice of categorizing this paper as 'methodological development' seems to be in question, as several reviewers noted that there is limited technological novelty, but more an application of interesting methodological development to an important problem of privacy and identity inference. I would recommend to the authors that they switch the type of paper to 'both' or 'application'.

Most of the reviewers had concerns about the experiments, especially emphasizing aspects like statistical significant, improving (or at least characterizing) the reality of the scenarios tested, adapting the claims made in the paper to match the results, and generalizability of the conclusions (among a bunch of smaller suggestions relating to clarity of figures and so on). The authors seem to have addressed these issue sin the responses (and all reviewers improved to or kept accept-like scores) ,  but I would encourage the authors to still parse these experiments carefully and in the future do these properly from the start -- some of these issues are fundamental in the field. For example, including some sort of notion of significant (e.g. via standard deviation of attacks) is crucial to a first submission.

Overall, I appreciate the overall thoroughness of the response to the authors -- they are both well organized and thorough in explanation, experiments and revision.

**Paper Type:**

both

---

### Decision · Program_Chairs · 2021-03-31

Accept